# Incidence of Occult Hepatitis B Infection (OBI) and hepatitis B genotype characterization among blood donors in Cameroon

**Macqueen Ngum Mbencho**[1,2‡], **Nourhane Hafza**[1‡], **Le Chi Cao**[1,3], **Victorine Ndiwago Mingo**[2], **Eric A. Achidi**[4], **Stephen Mbigha Ghogomu**[2], **Thirumalaisamy P. Velavan**[1,5,6]*

1 Institute of Tropical Medicine, University of Tübingen, Tübingen, Germany, 2 Molecular and Cell Biology Laboratory, University of Buea, Buea, Cameroon, 3 Department of Parasitology, Hue University of Medicine and Pharmacy (HUMP), Hue University, Hue, Vietnam, 4 Faculty of Sciences, University of Buea, Buea, Cameroon, 5 Vietnamese-German Center for Medical Research (VG-CARE), Hanoi, Vietnam, 6 Faculty of Medicine, Duy Tan University, Da Nang, Vietnam

‡ MNM and NH are share first authors on this work.
* t.velavan@uni-tuebingen.de

**Data Availability Statement:** All data generated or analysed during this study are included in this article. A total of 14 successfully sequenced HBV

## Abstract

### Background

Occult hepatitis B infection (OBI) is characterized by the presence of hepatitis B virus (HBV) DNA at low levels in serum (<200 IU/mL) with a negative hepatitis B surface antigen (HBsAg) test. OBI remains a major challenge to blood safety, particularly in HBV-endemic regions like Cameroon, where HBV detection relies solely on HBsAg testing. This cross-sectional study aimed to investigate the actual incidence and genotype characteristics of OBI in Cameroonian blood donors.

### Methods

Between March and June 2023, samples were collected from 288 HBsAg-negative blood donors aged 18 to 55 years and analysed for antibodies against the HBV core (anti-HBc) and surface antigens (anti-HBs). Following DNA extraction from the serum samples, qualitative nested PCR and quantitative real-time PCR were used to detect HBV viral DNA and viral load respectively. For positive samples, sequencing of a fragment of the *S* gene was performed to identify the circulating HBV genotypes.

### Results

The findings revealed that 58% (n = 167/288) of blood donors tested positive for anti-HBc, 29% (n = 83/288) tested positive for anti-HBs, and 26% (n = 75/288) being positive for both anti-HBc and anti-HBs. Occult hepatitis was confirmed in 4.5% of the blood donors, all of whom belonged to either HBV genotypes *A* or *E*, which are predominant in Cameroon. The amino acid substitution *sA184V* associated with HBsAg detection failure in genotype E was observed in 70% of OBI sequences, and the HBsAg immune escape variants (*sT131N* and *sS143L*) implicated in OBI were also observed. The mutation *rtN139K* in the reverse

DNA positive samples were submitted to the NCBI GenBank database (https://www.ncbi.nlm.nih.gov/genbank/) with accession numbers PP746847-PP746860 (n=14).

**Funding:** The project was funded by the PAN-ASEAN Coalition for Epidemic and Outbreak Preparedness (PACE-UP; German Academic Exchange Service (DAAD) Project ID: 57592343). The funder had no role in the study design, data collection and analysis, decision to publish, or preparation of the manuscript.

**Competing interests:** The authors declare no conflict of interest.

transcriptase (RT) domain of the overlapping HBV polymerase (*P*) gene was present in 17% of OBI-positive sequences of genotype *E*, likely contributing to masking HBsAg secretion.

## Conclusion

The results suggest a considerable risk of transfusion-transmitted HBV in this region. Therefore, to ensure blood safety, nucleic acid testing (NAT) is recommended, as relying solely on HBsAg assays is insufficient to eliminate this risk.

## Introduction

Hepatitis B, caused by the hepatitis B virus (HBV), poses a significant health challenge, especially in Africa. Despite the World Health Organization (WHO) aiming to eliminate viral hepatitis as a public threat by 2030 [1], many African countries, including Cameroon, continue to struggle with high prevalence rates. Cameroon reports a pooled HBV prevalence of 11% [2]. The introduction of HBV vaccination in 2005 as part of the Expanded Program on Immunization (EPI) for infants (administered at 6, 10, and 14 weeks of age), aimed to address this problem. Before its implementation, the seroprevalence rates were as high as 20% among school-aged children [3], and 14% in the adult population [4]. Moreover, the vaccine is not provided free for children born before 2005, who would now be adult chronic carriers if infected.

Among other viral infections such as hepatitis C (HCV) and Human Immunodeficiency Virus (HIV), HBV is routinely screened by serology to ensure the safety of blood transfusions. However, despite testing negative for the hepatitis B surface antigen (HBsAg), there remains a residual risk of HBV transmission, wherein detectable viral DNA in the liver and/or blood, typically <200IU/mL, contributes to the risk of transfusion-transmitted infection. This phenomenon is referred to as occult hepatitis B (OBI) [5], which significantly contributes to silent HBV transmissions. The molecular basis of OBI lies in the long-term persistence of a stable replication-competent episomal HBV covalently closed circular DNA (cccDNA) within the nucleus of infected hepatocytes [5]. The mechanisms underlying OBI are multifactorial, including both host and viral factors that contribute to the suppression of viral replication, thereby keeping the virus under control. However, immunosuppressed individuals, such as patients undergoing chemotherapy, hemodialysis, or those with HIV who receive transfusions containing HBV, are at risk of reactivation of the transcriptional activity of the viral cccDNA, leading to a full-blown overt infection that often progresses to liver cirrhosis and hepatocellular carcinoma [6].

In developed countries, nucleic acid tests (NAT) for screening blood donations are increasingly used to reduce the risk of OBI [7]. In contrast, other countries utilize antibodies to the HBV core protein (anti-HBc) as a surrogate marker. However, OBI remains largely unrecognized in low-to-middle-income countries like Cameroon, where blood donor selection relies solely on serological testing of HBsAg. Data on the prevalence of OBI in Cameroon is limited. In 2019, a study (in Yaoundé, Cameroon) reported an OBI incidence of 1.1% among 522 anti-HBc reactive donor samples [8], while another study in 2021 reported an OBI residual risk of 1.6% in seropositive donors from the same region [9]. These studies have not considered seronegative donors despite hypothesis suggesting that OBI development involves stronger suppression of viral replication and HBsAg expression. For instance, mutations in the *PreS/S* regions might alter HBsAg expression, secretion, and antigenicity, thus inhibiting the production of anti-HBs [10].

This study investigated the OBI incidence, defined by the presence of hepatitis B viral DNA in the serum, associated viral factors, and circulating HBV genotypes in the blood donor population of the Southwest Region of Cameroon who tested negative for HBsAg.

## Materials and methods

### Ethics statement

This study was approved by the Institutional Review Board of the University of Buea, Cameroon (Ethics approval number: 2022/1849-10/UB/SG/IRB/FHS) and the University of Tübingen for the project 'Molecular surveillance of hepatitis E and Occult hepatitis B in the Cameroonian population' (Ethics approval number: 379/2023B02). In addition, administrative clearance was obtained from the Southwest Regional Delegation of Public Health (Approval number: 84/MPH/SWR/RHL/DO/03/2023). Signed informed consent was obtained from all study participants prior to enrolment.

### Study cohort and sampling

Between March and June 2023, 288 consented, healthy blood donors aged 18 to 55 years were recruited from the Buea Regional Hospital blood bank in Cameroon. Demographic data, vaccination status, and blood donation history were recorded using a structured questionnaire in this cross-sectional study. After meeting the eligibility criteria, which included being HBsAg-negative, having no medical history of chronic disease, and testing negative for HIV, HCV, and syphilis, 3 mL of blood was collected from each donor. The eligibility criteria for blood donation included being aged 18 to 65 years, weighing at least 50 kg, being in good general health, free from acute illness, with normal blood pressure and pulse rates, and having adequate hemoglobin levels as determined during pre-donation screening. Individuals with chronic diseases, those on medication, and pregnant women are not eligible to donate. The HBsAg status, HIV, HCV, and Syphilis was determined using rapid test kits and ELISA (Fortress Diagnostics, Antrim, United Kingdom). The HBsAg assay used a sensitivity of 99.75% and a specificity of 99.87%, while the anti-HCV assay had a sensitivity of 99.79% and a specificity of 99.55%. Additionally, the HIV (Ag/Ab) assay had an analytical sensitivity for p24 antigen of 5 pg/mL, and the *Treponema pallidum* antibody rapid test kit showed a sensitivity of 100% and a specificity of 99.7%.

### Screening of HBV serological markers

All 288 blood donor sera were screened for HBV serological markers anti-HBs (Monolisa™ anti-HBs PLUS, Bio-Rad, Hercules, CA, USA) and anti-HBc (Monolisa™ anti-HBc PLUS, Bio-Rad, Hercules, CA, USA) using ELISA procedures, following manufacturer's instructions. The absorbance was measured on a CLARIOstar microplate reader (BMG Labtech, Ortenberg, Germany). The anti-HBs positivity was defined as a titer value >10 mIU/mL and the assay had a sensitivity of 99.2% and a specificity of 99.4%. The anti-HBc assay (qualitative) had a sensitivity of 99.53% and a specificity of 99.9%.

### Qualitative and quantitative detection of HBV

HBV DNA was extracted from 200μl of serum using the QIAamp Viral DNA mini kit (Qiagen GmbH, Hilden, Germany). For HBV DNA qualitative detection, HBV-specific nested-PCR targeting a highly conserved overlapping *S/P* region (332 bp) was conducted as previously described [11–13]. PCR reactions (25μL) consisted of 1xPCR buffer, 0.2 mM dNTPs, 0.4 μM primers (S1 Table), and 1U Taq DNA Polymerase (Qiagen GmbH, Hilden, Germany).

Thermal cycling parameters include initial denaturation at 94°C for 5 mins, followed by 35 cycles of denaturation (94°C, 30s), annealing (55°C for outer, 54°C for inner, 30s), and extension (72°C, 30s), with a final extension at 72°C for 5 mins. Controls (positive: HBV plasmid DNA; negative: master mix) were incorporated to validate PCR products. The nested PCR detection limit for HBV DNA was approximately 2.5 copies per reaction, equivalent to 30 to 40 copies/mL. Amplicons were visualized by agarose gel electrophoresis. PCR positives were purified using (Thermo Fisher Scientific, Waltham, MA, USA) and sequenced with the Big Dye™ Terminator v.3.1 Cycle Sequencing Kit (Thermo Fisher Scientific, Waltham, MA, USA) on an Applied Biosystems 3130xl Genetic Analyzer (Applied Biosystems, Beverly, MA, USA) following the manufacturer's protocol.

For DNA quantification, real-time PCR was conducted using virus-specific primers and a Taqman probe that targeted a 90-bp fragment within the conserved region of the S gene in the HBV genome (position 182–271, GenBank number X75657), following a previously described protocol [14]. The SensiFAST™ one-step RT-PCR kit (Meridian Biosciences, Memphis, Tennessee, USA) was used on a LightCycler480-II (Roche, Mannheim, Germany). Each real-time PCR reaction was performed in 20 μL volume, containing 0.8 μL each of 10 μM forward primer (HBV-61) and reverse primer (HBV-62), 2X real-time-PCR master mix (10 μL), 0.3 μL probe (HBV TM-05) (S1 Table), with 15–20 ng (5 μL) of DNA. Cycling conditions included an initial denaturation at 95°C for 5 mins, followed by 45 cycles of 95°C for 10 seconds and 60°C for 34 seconds. The limit of detection for the real-time PCR used to quantify HBV viral load was 25 IU/mL. This limit was established using a control plasmid with a concentration of 10^6 copies/μL, serially diluted tenfold. The linear equation relating Ct values to log (copies/μL) was used to calculate viral loads. The highest Ct value detected, which was 40, corresponded to 25 IU/mL.

## Phylogenetic and mutation analysis of HBV sequences

The HBV-specific gene sequences obtained were trimmed in Seqman version 6.1 (DNASTAR, Lasergene, USA) and the resulting consensus sequences were manually verified. Alignment was performed using MAFFT version 7.0 using the G-INS-I model [15]. For the phylogenetic tree reconstruction, a combined set of 28 representative reference sequences was used consisting of 16 sequences specific to HBV genotypes *A-H* retrieved from the NCBI GenBank (https://www.ncbi.nlm.nih.gov/genbank/): Genotypes *A* (M57663, AF090842), *B* (D23678, D50522), *C* (D23680, M38636), *D* (AF151735, X02496), *E* (AB032431, X75657), *F* (X69798, AY090455), *G* (AB064313, AF160501), and *H* (AB059659, AY090454) along with an additional 12 sequences identified through an NCBI BLAST (https://blast.ncbi.nlm.nih.gov/Blast.cgi) search for the HBV DNA positive sequences: Genotypes *A* (LC513655, MH213535, MH580639), and *E* (KY494007, MH253771, MK840530, MK174170, KU702933, DQ060826, MF772360, MN507847, KU522302). The phylogenetic tree was reconstructed using MEGA version 11 [16] employing the Maximum Likelihood method using the Kimura 2 parameter plus Gamma Distribution model (K2+G). The statistical robustness and reliability of the branching order were confirmed via bootstrapping with 1000 replicates. The resulting phylogenetic tree was annotated and visualized using the online tool iTOL v6 (https://itol.embl.de/) [17]. Mutation analysis was done on BioEdit version 7.2.6 (https://bioedit.software.informer.com/7.2/) using genotype-matched references for the identified genotypes (*A* and *E*). The representative OBI sequences obtained in this study have been deposited in the NCBI GenBank database (https://www.ncbi.nlm.nih.gov/genbank/) and can be retrieved with accession numbers PP746847-PP746860 (n = 14).

### Data analysis

Statistical analysis was performed using R version 4.0. The median age of the blood donors was determined as the median value in the ordered age distribution, with variability described as interquartile range (IQR). The incidence of OBI was calculated as the proportion of HBsAg-negative blood donors who tested positive for HBV DNA and with a viral load not exceeding 200 IU/mL. Viral load in positive samples was quantified using real-time PCR, with results expressed in IU/mL based on a standard curve generated from a serially diluted control plasmid with a concentration of 10^6 copies/μL.

## Results

### Demographic characteristics of the study cohort

Of the 288 blood donors sampled, males dominated, accounting for 97% of the study cohort (n = 279). The median age of donors was 30 years (IQR; 24–38) and the dominant age group was 21–30 years (48%; n = 137) followed by 31–40 years old (32%; n = 91). Majority of donors had a donation history (79%; n = 227) and only 3% (n = 9) had been vaccinated for HBV.

### Serological profiles

The serology results for HBV showed that 58% (n = 167/288) were positive for anti-HBc and 29% (n = 83/288) positive for anti-HBs. Of these, 26% (n = 75/288) tested positive for both antibodies and 39% (n = 113) tested negative for both anti-HBc and anti-HBs antibodies (Fig 1).

### HBV DNA detection

HBV DNA was detected in 14 out of 288 samples (5%) by nested PCR. The median age of the HBV DNA-positive blood donors was 24 years (range: 18–41). Among these samples, four had detectable viral loads measured by qPCR, ranging from <25 to 2593 IU/mL (Table 1). Three of those with detectable viremia were positive for anti-HBc, while one was positive for anti-HBs but not anti-HBc. Of all samples tested positive for HBV DNA, 93% remained negative for anti-HBs (Table 1). Following the consensus definition of OBI, one HBV DNA positive sample (BD229) was considered 'false' OBI since its viral load is > 200IU/mL. Therefore, the incidence of OBI in this population is 4.5%.

### OBI genotyping and genetic variation

Sequences amplified from the *S/P* region of HBV were aligned with the reference sequences from the NCBI genotyping tools and the HBV database (HBVdb). BLAST searches identified 13 sequences as genotype E and one as genotype A (Table 1). Phylogenetic analysis with 28 reference sequences corroborated these findings (Fig 2). Using reference genomes obtained from NCBI tool (https://www.ncbi.nlm.nih.gov/nuccore/; M57663 and AF090842 for genotype *A*; AB032431 and X75657 for genotype *E*), the *S* and RT domains of OBI-positive sequences were analyzed. Nonsynonymous substitutions in the amino acids were found in the S protein's "α" determinant region (aa 124–147) within the major hydrophilic region (MHR), and at the reverse transcriptase (RT) domain of the *P* gene (Table 2, S1 and S2 Figs). Notably, the *sA184V* substitution was present in 70% of the OBI sequences, and *sY206C* was also observed in 62% of sequences, both located outside the "α" determinant and MHR (aa 99–169) regions. Within the "α" determinant, substitutions *sT131N* and *sS143L* were observed, while *sK122R* and *sK160R* were found in the major hydrophilic region. The mutations *rtW153R*, *rtI163V*, *rtI122V*, *rtN139K*, and *rtF151L* were present in the RT domain. The sample with a high viral

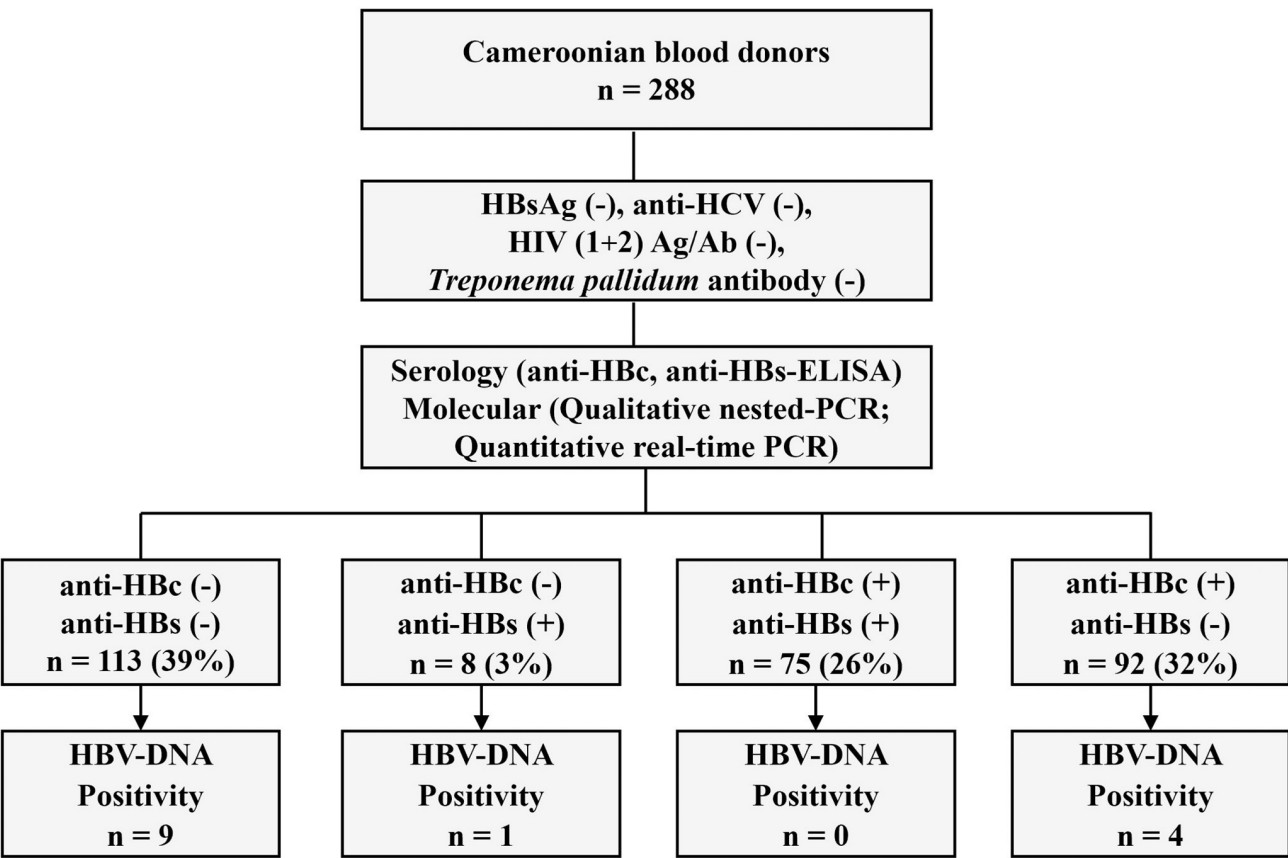

**Fig 1. Study design and summary of results.** A total of 288 HBsAg-negative blood donor serum were analyzed by Enzyme Linked Immunosorbent Assay (ELISA) for anti-HBc and anti-HBs, and subsequently by nested PCR for HBV DNA in the highly conserved *S/P* region of the HBV genome.

**Table 1. Characteristics of OBI blood donors in terms of seropositivity, viral load, and hepatitis B virus (HBV) genotypes.**

| Sample ID | anti-HBc | anti-HBs | HBV PCR | HBV qPCR | Viral load (IU/mL) | HBV Genotypes |
|---|---|---|---|---|---|---|
| BD072 | - | - | + | - | undetectable | *E* |
| BD078 | - | - | + | - | undetectable | *E* |
| BD092 | - | - | + | - | undetectable | *E* |
| BD118 | - | - | + | - | undetectable | *E* |
| BD147 | + | - | + | + | 81 | *A* |
| BD157 | - | - | + | - | undetectable | *E* |
| BD163 | - | - | + | - | undetectable | *E* |
| BD172 | - | - | + | - | undetectable | *E* |
| BD174 | - | - | + | - | undetectable | *E* |
| BD183 | - | - | + | - | undetectable | *E* |
| BD211 | + | - | + | - | undetectable | *E* |
| BD225 | + | - | + | + | <25 | *E* |
| BD229 | + | - | + | + | 2593 | *E* |
| BD284 | - | + | + | + | 31 | *E* |

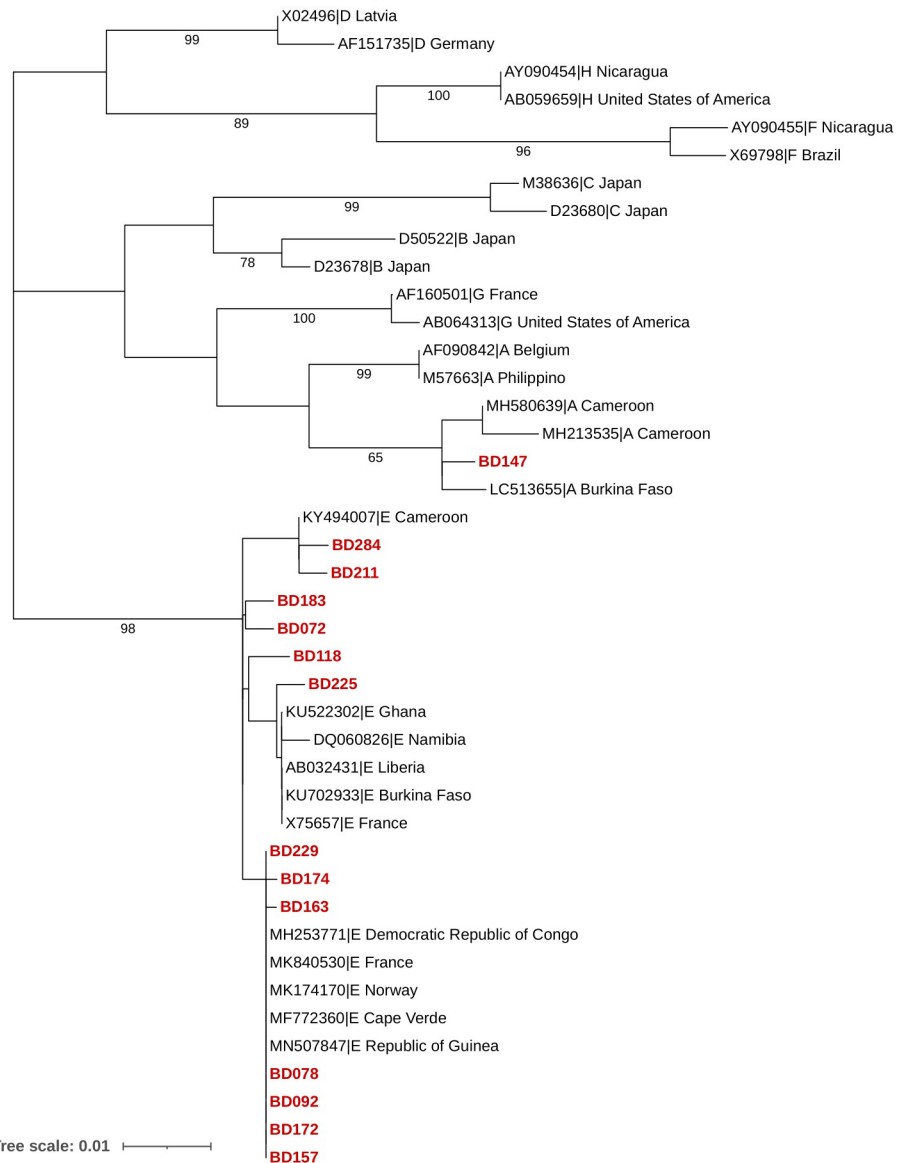

**Fig 2. Reconstructed phylogenetic tree of the *S/P* gene of the HBV genome.** The Maximum Likelihood method was employed for the reconstruction of the phylogenetic tree, and the evolutionary distances were computed using the Kimura 2-parameter plus Gamma Distribution (K2+G) model with 28 representative sequences (16 specific references for HBV genotypes A-H, and 12 sequences obtained through NCBI BLAST search; https://blast.ncbi.nlm.nih.gov/ Blast.cgi). The OBI sequences from blood donors in this study clustered in the HBV genotypes A and E. The numbers at the nodes indicate bootstrapping values as a percentage of 1000 replicates. Only bootstrap values > 60 are shown. The OBI sequences are highlighted in red. BD—Blood donor.

load (2593 IU/mL), exhibited mutation *sA184V* consistent with findings in 70% of all samples tested.

## Discussion

Despite the availability of vaccines, occult hepatitis B infection (OBI) remains a significant challenge to HBV elimination. In this study, we found a 4.5% incidence of OBI in the population of Buea, Southwest region of Cameroon. This raises a significant concern, particularly for

**Table 2. Mutations observed in the hepatitis B virus (HBV) (*S*) gene and in the reverse transcriptase (RT) domain of the polymerase (*P*) gene.**

| HBV genome | Amino acid substitution | HBV Genotypes |
|---|---|---|
| 'S' gene that encodes the Hepatitis B surface antigen (HBsAg) | sK122R | A |
| | sA194V | |
| | sT131N | E |
| | sS143L | |
| | sK160R | |
| | sA184V | |
| | sY206C | |
| 'P' gene that encodes the viral DNA polymerase, essential for the replication of the viral genome. | rtW153R | A |
| | rtI163V | |
| | rtI122V | E |
| | rtN139K | |
| | rtF151L | |

immunocompromised recipients, such as those undergoing hemodialysis, living with HIV, receiving immunosuppressive therapies, or suffering from liver disease. In these patients, there is a heightened risk for reactivation of HBV replication which could lead to acute hepatitis B infection [5].

Furthermore, there was a high seroprevalence of anti-HBc, observed at 58%. The OBI carriage rate reported in this study is notably higher compared to the prevalence of 1.1% [8] and 1.6% [9] previously reported in the Centre Region of Cameroon, where anti-HBc seroprevalence among donors was 49%. A higher OBI prevalence (17%) has been reported in neighboring Nigeria [12]. The risk of HBV transmission from OBI is high in low- and middle-income countries due to the lack of anti-HBc testing and NAT. In contrast, developed countries use NAT for blood screening, significantly reducing the residual risk of HBV transmission [7]. The prevalence of OBI among blood donors typically reflects the endemicity of HBV, ranging from 0.06% in low-endemicity countries to 0.98% in high-endemicity countries [18]. Additionally, 64% of the OBI samples in this study were seronegative (negative for any HBV antibodies). Seronegative individuals can account for up to 20% of all OBI cases [5], suggesting that selective testing only seropositive donors for HBV DNA poses a risk for HBV transmission.

Variations in OBI prevalence may arise from socioeconomic factors, regional differences, and the criteria used for detecting OBI among seropositive and seronegative donors. All identified OBI carriers in this study were males, reflecting a common gender imbalance in donor populations in developing countries, where males typically outnumber females. This disparity complicates the assessment of OBI prevalence in relation to gender [19]. Notably, 97% of our study cohort was not vaccinated against HBV, including the 4.5% of individuals with OBI. This contrasts with the high vaccination coverage rate of 99% in the infant population [20] in Cameroon and highlights the lower vaccination coverage in the general adult population. Previous studies have shown that genotypes A and E co-circulate in Cameroon, with reported mixed infections and recombination between genotypes A and E. In our study, the majority of OBI sequences (93%) were identified as genotype *E*, which is commonly found in Cameroon. Only one sequence belonged to genotype *A*. Genotype *E* predominates across Western Africa, from Mauritania to Namibia [8]. Conversely, HBV genotype *A* has been notably prevalent in specific Cameroonian populations, including Pygmies, Bantus, and HIV-infected individuals [21, 22].

The mechanisms underlying OBI are complex and include variations in the HBV genome that can result in OBI. For instance, the "α" determinant region (aa 124–147) within the major hydrophilic region (MHR, aa 99–169) of the S protein is a mutational hotspot. Mutations here affect the expression, antigenicity, and immunogenicity of HBsAg, leading to detection failures by commercial assays [23]. Notably, the amino acid residues at positions 122 and 160 of the *S* gene (arginine, R or lysine, K) determine the serotypes *d/y* and *w/r*, respectively [24]. Specifically, R at position 122 defines subtype *y*, while K defines subtype *d* [24]. Substitutions at these positions have been linked to immune escape of HBsAg [25]. In this study, the *sK122R* mutation, which indicates a serotype change from *d* to *y*, was observed in OBI genotype *A*. This mutation has been significantly associated with OBI and is known to contribute to reduced HBsAg secretion, as shown in *in vitro* studies [10, 26]. Additionally, we observed the *sK160R* mutation (change from subtype *w* to *r*) in OBI individuals with genotype *E*, which has been demonstrated to reduce extracellular HBsAg expression in OBI individuals [27].

The *sT131N* mutation in the "α" determinant region was observed in 15% of OBI individuals with genotype *E*. This mutation has also been found in OBI blood donors with genotype *C* in a Chinese population [28] and is associated with low HBsAg concentrations (<100 IU/mL) [29] and reduced HBsAg antigenicity, contributing to immune escape [30]. The *sS143L* mutation found in genotype *E* in this study is a typical HBsAg escape mutant and has been detected more frequently in Italian OBI blood donors of genotype *D* [31]. Moreover, the *sA184V* amino acid substitution (70% in genotype *E*) observed in this study was associated with impaired HBsAg detection in genotype *E* HBV/HIV-infected Nigerian population [32]. Also, *sY206C* observed in 62% of OBI sequences is located in the C-terminus of HBsAg known to be involved in virion and/or HBsAg secretion [30]. In a previous study, this variant was significantly associated with low HBsAg and serum HBV-DNA levels [33].

OBI sequences revealed several documented RT mutations. Among these, *rtN139K* (17%, genotype *E*) is common in treatment-naïve patients in Asia [34] and significantly associated with progression to HCC [35]. The primers targeted the overlapping S/P region of the HBV genome, spanning nucleotides 455 to 786, which includes the Reverse Transcriptase (RT) domain of the Polymerase gene—a region associated with resistance to nucleoside/nucleotide analogues (NAs). Upon analysis using geno2pheno and the HBV database, no well-characterized drug-resistance mutations were detected. However, some of the mutations we identified, such as rtW153R and rtI163V, have been associated with resistance to Adefovir and Lamivudine in previous studies investigating RT mutations in both treated and untreated individuals [36, 37]. The *rtI122V* mutation (8%) detected in our study was also reported in the Indonesian population as a putative drug resistance mutant [38]. While amino acid substitutions within the RT domain may affect the efficiency of viral replication and HBsAg secretion, their direct association with OBI remains unclear.

In one sample with a high viral load (>2000 IU/mL), one mutation (*sA184V*) was observed which also occurred in nine other OBI sequences. Despite the high viremia, qualitative ELISA HBsAg assay used in the blood bank, which has a detection limit of 0.5 ng/mL, failed to detect it, suggesting a potential laboratory misdiagnosis. This study has several limitations. It is cross-sectional, and all blood donors were male, indicating a gender bias in the study cohort. Additionally, individuals with clinically confirmed HBV infection from this population were not included, preventing comparative analysis.

## Conclusion

Our study highlights that relying solely on routine HBsAg testing is inadequate to prevent HBV transmission through transfusions in endemic regions. We recommend supplementing

HBsAg testing with nucleic acid amplification testing (NAT) and/or anti-HBc testing as surrogates for detecting OBI.

## Supporting information

**S1 Table. Primers and probes used for HBV DNA detection.**
(PDF)

**S1 Fig. HBV surface protein (S) alignment.** Surface protein alignment of OBI-positive sequences with reference sequences for genotypes *A* and *E* (reference genomes: M57663|A, AF090842|A, X75657|E, and AB032431|E written in red). The alignment shows amino acids 100 to 210 of the S protein. A dot represents homology in amino acids as observed in the reference sequence; only those with amino acid substitutions are illustrated. Bold squares indicate amino acid substitution for genotype *A* and dash lines show amino acid substitutions for genotype *E*.
(PDF)

**S2 Fig. Reverse transcriptase (RT) alignment of the polymerase (*P*) gene.** The alignment of the RT region of the *P* gene of OBI-positive sequences with reference sequences for genotypes *A* and *E* (reference genomes: M57663|A, AF090842|A, X75657|E, and AB032431|E, written in red). The alignment shows amino acids 110 to 220 of the RT domain. A dot represents homology in amino acids as observed in the reference sequence; only those with amino acid substitutions are illustrated. Bold squares indicate amino acid substitution for genotype *A* and dash lines show amino acid substitutions for genotype *E*.
(PDF)

**S1 File. Information sheet, consent form, and questionnaire.**
(PDF)

## Acknowledgments

The authors thank all blood donors and the entire blood bank team of the Buea Regional Hospital for their cooperation. The authors would like to thank Ms. Le Thi Kieu Linh for supporting sequencing procedures.

## Author Contributions

**Conceptualization:** Eric A. Achidi, Stephen Mbigha Ghogomu, Thirumalaisamy P. Velavan.

**Data curation:** Macqueen Ngum Mbencho, Nourhane Hafza, Le Chi Cao.

**Formal analysis:** Macqueen Ngum Mbencho.

**Funding acquisition:** Thirumalaisamy P. Velavan.

**Investigation:** Macqueen Ngum Mbencho, Nourhane Hafza.

**Methodology:** Macqueen Ngum Mbencho, Nourhane Hafza, Le Chi Cao, Victorine Ndiwago Mingo.

**Project administration:** Thirumalaisamy P. Velavan.

**Resources:** Victorine Ndiwago Mingo, Eric A. Achidi, Stephen Mbigha Ghogomu.

**Software:** Macqueen Ngum Mbencho.

**Supervision:** Stephen Mbigha Ghogomu, Thirumalaisamy P. Velavan.

**Validation:** Macqueen Ngum Mbencho.

**Visualization:** Nourhane Hafza, Le Chi Cao.

**Writing – original draft:** Macqueen Ngum Mbencho.

**Writing – review & editing:** Nourhane Hafza, Thirumalaisamy P. Velavan.

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
