## [Decision Letter · Decision Letter 0]

8 Aug 2024

PONE-D-24-28215Prevalence and genotype characteristics of Occult Hepatitis B infection among blood donors in Cameroon: Implications for blood safetyPLOS ONE

Dear Dr. Velavan,

Thank you for submitting your manuscript to PLOS ONE. After careful consideration, we feel that it has merit but does not fully meet PLOS ONE’s publication criteria as it currently stands. Therefore, we invite you to submit a revised version of the manuscript that addresses the points raised during the review process.

We look forward to receiving your revised manuscript.

Kind regards,

Jason T. Blackard, PhD

Academic Editor

PLOS ONE

Journal Requirements:

2. We note that this data set consists of interview transcripts. Can you please confirm that all participants gave consent for interview transcript to be published?

If they DID provide consent for these transcripts to be published, please also confirm that the transcripts do not contain any potentially identifying information (or let us know if the participants consented to having their personal details published and made publicly available). We consider the following details to be identifying information:

- Names, nicknames, and initials

- Age more specific than round numbers

- GPS coordinates, physical addresses, IP addresses, email addresses

- Information in small sample sizes (e.g. 40 students from X class in X year at X university)

- Specific dates (e.g. visit dates, interview dates)

- ID numbers

Or, if the participants DID NOT provide consent for these transcripts to be published:

- Provide a de-identified version of the data or excerpts of interview responses

- Provide information regarding how these transcripts can be accessed by researchers who meet the criteria for access to confidential data, including:

a) the grounds for restriction

b) the name of the ethics committee, Institutional Review Board, or third-party organization that is imposing sharing restrictions on the data

c) a non-author, institutional point of contact that is able to field data access queries, in the interest of maintaining long-term data accessibility.

d) Any relevant data set names, URLs, DOIs, etc. that an independent researcher would need in order to request your minimal data set.

For further information on sharing data that contains sensitive participant information, please see: https://journals.plos.org/plosone/s/data-availability#loc-human-research-participant-data-and-other-sensitive-data

If there are ethical, legal, or third-party restrictions upon your dataset, you must provide all of the following details (https://journals.plos.org/plosone/s/data-availability#loc-acceptable-data-access-restrictions):

a. A complete description of the dataset

b. The nature of the restrictions upon the data (ethical, legal, or owned by a third party) and the reasoning behind them

c. The full name of the body imposing the restrictions upon your dataset (ethics committee, institution, data access committee, etc)

d. If the data are owned by a third party, confirmation of whether the authors received any special privileges in accessing the data that other researchers would not have

e. Direct, non-author contact information (preferably email) for the body imposing the restrictions upon the data, to which data access requests can be sent.

**Additional Editor Comments:**

This is a cross-sectional study of occult HBV conducted in blood donors in Cameroon.

Given the high burden of HBV in resource-limited settings, such studies are important.

The population size is relatively modest and the finding of 14 occult HBV infections is not surprising.

What region of the viral genome is targeted by the real-time PCR assay?  Were samples that tested negative for HBV DNA re-tested a second time?

By definition, occult HBV cases are HBsAg negative but HBV DNA positive.  Only 4 samples included in Table 1 actually have detectable HBV DNA by real-time PCR.

For the mutational analysis, what were the study sequences compared to?  Genotype-matched references?  How many? What was the definition of a mutation then?  Any mutation not detected in the reference set?  Above a certain threshold (say 10%)?

Figure 1:  the black background is distracting and should be removed.

Figure 2:  this should be redrawn as a phylogram such that branch lengths denote genetic distance.

How different are the various sequences at the nucleotide level?

What HBV genotypes have been found chronic HBV infections in Cameroon?

Does this region of S overlap with sites associated with drug resistance in P?  what drug resistance mutations were observed?

Reviewers' comments:

Reviewer's Responses to Questions

**Comments to the Author**

1. Is the manuscript technically sound, and do the data support the conclusions?

Reviewer #1: Partly

Reviewer #2: Partly

2. Has the statistical analysis been performed appropriately and rigorously? 

Reviewer #1: Yes

Reviewer #2: No

3. Have the authors made all data underlying the findings in their manuscript fully available?

Reviewer #1: Yes

Reviewer #2: Yes

4. Is the manuscript presented in an intelligible fashion and written in standard English?

Reviewer #1: Yes

Reviewer #2: Yes

5. Review Comments to the Author

**Reviewer #1:** This study is relevant to the transfusion field in Africa. However, some aspects need to be improved, especially the methodology.

Title

1.Is the sample sufficient to determine prevalence? I suggest changing the title.

Abstract

2.It should be made clear in the abstract that these are HBsAg-negative blood donors.

Introduction

3.In the last paragraph, the authors mention the incidence of occult Hepatitis B. I suggest alignment with the title and taking into account comment 1.

Methodology

4. The recruiting process of 288 HBsAg negative blood donors needs to be clarified.

From which population were they recruited (blood donation candidates or approved donation candidates)?

How many blood donors were included to obtain 288 HBsAg negatives? What was the frequency of HBsAg in these blood donors?

5. I suggest a general description of the criteria for blood donation mentioned in number 113.

6. Serological tests have a low sensitivity for HBV (HBsAg) screening compared to molecular testing. What strategies were used to reduce false negative results?

7. I suggest that you also indicate the sensitivity and specificity of the tests used for other serological markers.

8.What is the detection limit of the quantitative test used for HBV.

Results

9.Sample BD229 has a high viral load. Without confirmatory testing for HBsAg, we cannot exclude the possibility of it being a sample in an immunological window period or a false negative in the serological test.

Discussion

10. Despite this being part of the limitation, it is not clear why HBsAg-positive samples were excluded.

**Reviewer #2: **This study investigated the prevalence of OBI, which is defined by the presence of serum viral DNA for hepatitis B, associated viral factors, and the HBV genotype circulating in blood donors in southwestern Cameroon who tested negative for HBsAg and concluded that relying on routine HBsAg testing is insufficient to prevent HBV transmission through transfusions in endemic regions and recommend supplementing HBsAg testing with nucleic acid amplification testing (NAT) and/or anti-HBc testing as surrogates for detecting OBI

Abstract

• The abstract mentions only anti-HBs and anti-HBc testing. However, the HBsAg test was not included and was only mentioned in the methodology section. HBsAg tests should also be included.

Methods

• The researcher should describe the design of the study. Is this a cross-sectional study? If so, this needs to be explained

• The researcher should describe the study design. Is it a cross-sectional study? If so, it is necessary to explain this.

• Line 122: What is the titre value of the positive anti-HBc test? Give the value of the positive anti-HBc test, as the researcher have done for anti-HBc.

• Be consistent with the writing of the markers in some the researcher used both higher case letter and lower-case letters and it may be confusing to the reader e.g. Anti-HBs/ anti-HBs.

• Briefly describe the method (statistical analysis and calculations) used to calculate the prevalence, median age and viral load as the age in median, viral load in mean/ average were reported.

Results

• The HBsAg results are not reported, it is not clear if the researcher performed the HBsAg test, or it was previously tested prior to the study. If it was performed, this should be described as well.

• Lines 170 – 173: Anti-HBs and anti-HBc are reported twice. The anti-HBs as 29% and 3%; and anti-HBc as 58% and 32%.

• The results section needs to be revised. For example, 26% (n = 75) tested positive for both antibodies and 39% (n=113) were negative for both markers. When one subtracts 75 from 288 The researcher gets 213 and not 113. What is the status of the 100 samples. I suggest that the researcher revise these results.

• Line 184: Almost all tested positive remained negative for anti-HBs (13/14; 93%).

• Since the researcher reports the prevalence, the word “almost” should not be used and the researcher should provide the frequency in numbers or % only.

• Among these samples, four had 182 detectable viral loads measured by qPCR, ranging from <10 to 2593 IU/mL. The results report on the viral load, but not on the OBI. Mention how many samples are OBI. Also, clarify if the sample (BD299) with a viral load of 2593 IU/mL considered OBI. According to the OBI definition, the viral load is <200 IU/mL.

Phylogenetic tree

• Be consistent with writing figure 2 or Fig 2.

• The reference sequences are not written in the correct standard order, e.g., the correct way of writing ref. seq is genotype accession no|location. The description of the identity of the study sequences with the reference sequences is not clear, since the reference sequences are not written correctly.

• Under methods section, it is mentioned that the bootstrap of 1000 replicates was used to create the tree and that the nodes indicate indicated the bootstrap values. the bootstrap values are not included in the phylogenetic tree, so this tree does not represent a bootstrap phylogenetic tree. See lines 213-214.

• Phylogenetic analysis with 30 195 reference sequences corroborated these findings (Fig 2). The image has poor quality and resolution. I suggest that the researcher revise the construction of the phylogenetic tree.

Sequence analysis

• Since reference genomes (U87742 for 196 genotype A and AB091256 for genotype E) were used for constructing a phylogenetic tree. Is there any reason why they are excluded for mutation analysis comparison?

• Under S1 and S2- write the genotypes of the reference sequences AAC58023.3, BAC65109.1, AAM12945.1, BAC65108. Why did the researcher use different reference sequences for the S and RT regions? Combine the standard reference sequences in both alignments for the S and RT regions so that we know the genotypes from which these OBI mutations have been previously identified.

• The reference sequence on the top ‘AAC58023.3’ has a mutation 131N like the samples, which indicates it might not be a wild-type sequence because the other reference sequence ‘BAC65109.1’ below has wildtype base T131. Similarly, AAC58023.3 has T143 and “BAC65109.1 has S143.

• Is there a reason why Y206C is not reported on the results, as it shows to be a variant in the alignment (S1)

• Is there a reason for the separate alignment of the reference sequence ‘AAM12945.1’ and sample BD14 on top and with the other sequences below? I suggest The researcher combine the alignments and have all the alignment in one image/page. I suggest that the researcher include 2 or more standard references for both genotypes A and E.

Discussion

• In one sample with a high viral load (>2000 IU/mL), two mutations (sA184V and rtH125Q) 280 were observed that also occurred in other OBI sequences. On the basis of this statement above, is this sample having OBI or not. The load is high for it to be reported as OBI. The detection limit of 0.5 282 ng/mL is quite low to fail in the detection of HBsAg suggesting that the was nothing wrong with the HBsAg testing. The same sample with a high viral load has mutations that are different from the other samples, and the researcher did not discuss these differences and what could have impacted on this type of results.

• The researcher identified the mutations associated with drug resistance in the RT region; however, the researcher did not discuss how they impact on the OBI or the overlapping S region. This highlights another limitation of the study, which is the lack of information on the treatment history of the patients.

• The significance of the study results is not well discussed; the researcher needs to discuss the clinical implications of the OBI prevalence and mutations either towards e.g. diagnosis or treatment.

6. PLOS authors have the option to publish the peer review history of their article (what does this mean?). If published, this will include your full peer review and any attached files.

Reviewer #1: No

Reviewer #2: No

---

## [Author Response · Author response to Decision Letter 0]

27 Aug 2024

The rebuttals had been uploaded as a separate file

---

## [Decision Letter · Decision Letter 1]

2 Oct 2024

Incidence of Occult Hepatitis B Infection (OBI) and hepatitis B Genotype Characterization Among Blood Donors in Cameroon

PONE-D-24-28215R1

Dear Dr. Velavan,

We’re pleased to inform you that your manuscript has been judged scientifically suitable for publication and will be formally accepted for publication once it meets all outstanding technical requirements.

Kind regards,

Jason T. Blackard, PhD

Academic Editor

PLOS ONE

Additional Editor Comments (optional):

None

Reviewers' comments:

Reviewer's Responses to Questions

**Comments to the Author**

1. If the authors have adequately addressed your comments raised in a previous round of review and you feel that this manuscript is now acceptable for publication, you may indicate that here to bypass the “Comments to the Author” section, enter your conflict of interest statement in the “Confidential to Editor” section, and submit your "Accept" recommendation.

Reviewer #1: All comments have been addressed

Reviewer #2: All comments have been addressed

2. Is the manuscript technically sound, and do the data support the conclusions?

Reviewer #1: Yes

Reviewer #2: Yes

3. Has the statistical analysis been performed appropriately and rigorously? 

Reviewer #1: Yes

Reviewer #2: Yes

4. Have the authors made all data underlying the findings in their manuscript fully available?

Reviewer #1: Yes

Reviewer #2: Yes

5. Is the manuscript presented in an intelligible fashion and written in standard English?

Reviewer #1: Yes

Reviewer #2: Yes

6. Review Comments to the Author

Reviewer #1: (No Response)

Reviewer #2: The authors have addressed all the reviewer's comments satisfactorily and the manuscript should be accepted

7. PLOS authors have the option to publish the peer review history of their article (what does this mean?). If published, this will include your full peer review and any attached files.

Reviewer #1: No

Reviewer #2: No

---

## [Editor Report · Acceptance letter]

7 Oct 2024

PONE-D-24-28215R1 

PLOS ONE

Dear Dr. Velavan, 

I'm pleased to inform you that your manuscript has been deemed suitable for publication in PLOS ONE. Congratulations! Your manuscript is now being handed over to our production team.

Kind regards, 

on behalf of

Dr. Jason T. Blackard 

Academic Editor

PLOS ONE